**www.cambridge.org/qrd**

# Sequence – dynamics – function relationships in protein tyrosine phosphatases

Rory M. Crean[1] , Marina Corbella[1,2] , Ana R. Calixto[1,3] , Alvan C. Hengge[4] and Shina C. L. Kamerlin[1,5]

[1]Department of Chemistry – BMC, Uppsala University, Uppsala, Sweden; [2]Departament de Química Inorgànica i Orgànica (Secció de Química Orgànica) & Institut de Química Teòrica i Computacional (IQTCUB), Universitat de Barcelona, Barcelona, Spain; [3]LAQV, REQUIMTE, Departamento de Química e Bioquímica, Faculdade de Ciências, Universidade do Porto, Porto, Portugal; [4]Department of Chemistry and Biochemistry, Utah State University, Logan, UT, USA and [5]School of Chemistry and Biochemistry, Georgia Institute of Technology, Atlanta, GA, USA

## Research Article

**Keywords:**
empirical valence bond; enzyme evolution; loop dynamics; molecular simulations; protein tyrosine phosphatases

**Corresponding authors:**
Alvan C. Hengge and S. C. L. Kamerlin;
Emails: alvan.hengge@usu.edu;
skamerlin3@gatech.edu

## Abstract

Protein tyrosine phosphatases (PTPs) are crucial regulators of cellular signaling. Their activity is regulated by the motion of a conserved loop, the WPD-loop, from a catalytically inactive open to a catalytically active closed conformation. WPD-loop motion optimally positions a catalytically critical residue into the active site, and is directly linked to the turnover number of these enzymes. Crystal structures of chimeric PTPs constructed by grafting parts of the WPD-loop sequence of PTP1B onto the scaffold of YopH showed WPD-loops in a wide-open conformation never previously observed in either parent enzyme. This wide-open conformation has, however, been observed upon binding of small molecule inhibitors to other PTPs, suggesting the potential of targeting it for drug discovery efforts. Here, we have performed simulations of both enzymes and show that there are negligible energetic differences in the chemical step of catalysis, but significant differences in the dynamical properties of the WPD-loop. Detailed interaction network analysis provides insight into the molecular basis for this population shift to a wide-open conformation. Taken together, our study provides insight into the links between loop dynamics and chemistry in these YopH variants specifically, and how WPD-loop dynamic can be engineered through modification of the internal protein interaction network.

## Introduction

Protein tyrosine phosphatases (PTPs) are a broad superfamily of enzymes, which are crucial components of multiple cellular signaling pathways (Gurzov *et al.,* 2015). As a result, PTPs are implicated in several diseases, most notably multiple forms of cancer (Östman *et al.,* 2006), making them important (albeit elusive) drug targets (Barr, 2010; Mullard, 2018; Köhn, 2020).

PTPs share similar core structural features of a central parallel β-sheet, flanked by α-helices (Wang *et al.,* 2003). This core includes a crucial β-loop-α PTP signature motif, the central loop of which, the so-called 'P-loop', is critical for phosphate binding and catalysis. Specifically, the P-loop motif ($HCX_5R$) contains a cysteine nucleophile that initiates the catalytic cycle (Figure 1*a*), as well as a conserved arginine that positions the substrate and stabilizes the transition state. A second conserved loop, the WPD-loop, undergoes substantial (~10 Å) conformational transitions between catalytically inactive closed and catalytically active open conformations. In doing so, the WPD-loop optimally positions a key aspartic side chain (D356 in YopH) in the active site where it can act as an acid–base catalyst during the two-step cleavage/ hydrolysis mechanism common to all PTPs (Zhang, 1998). Other proximal loops such as the Q- and E-loops are less conformationally mobile but also play important roles in catalysis (Figure 1*d*; Crean *et al.,* 2021).

Curiously, despite sharing common conserved core structures, nearly identical active sites, and common catalytic mechanisms with similar transition states, the turnover numbers among PTPs vary by several orders of magnitude (Moise *et al.,* 2018). This strongly suggests an important role for loop dynamics in regulating PTP activity, a hypothesis supported by NMR (Whittier *et al.,* 2013) and by computational studies (Crean *et al.,* 2021) of the human protein tyrosine phosphatase 1B (PTP1B), as well as the *Yersinia* virulence factor YopH ('*Yersinia* outer protein H'), two of the most studied PTPs to date (e.g., Westheimer, 1987; Tonks *et al.,* 1988; Zhang *et al.,* 1992; Zhang and Dixon, 1994; Hunter, 1995, 2000; Johnson *et al.,* 2002; Zhang, 2002; Haj *et al.,* 2003; Tonks, 2003; Östman *et al.,* 2006; Zhang and Zhang, 2007, among many others). Further experimental and computational work has suggested that loop sequence significantly impacts the conformational dynamics of the loop, which can, in turn, affect both the activity and the pH dependency of catalysis (Shen *et al.,* 2021).

There has been a recent explosion of interest in the role of loop dynamics in enzyme evolution, and the successful application of loop manipulation to protein engineering

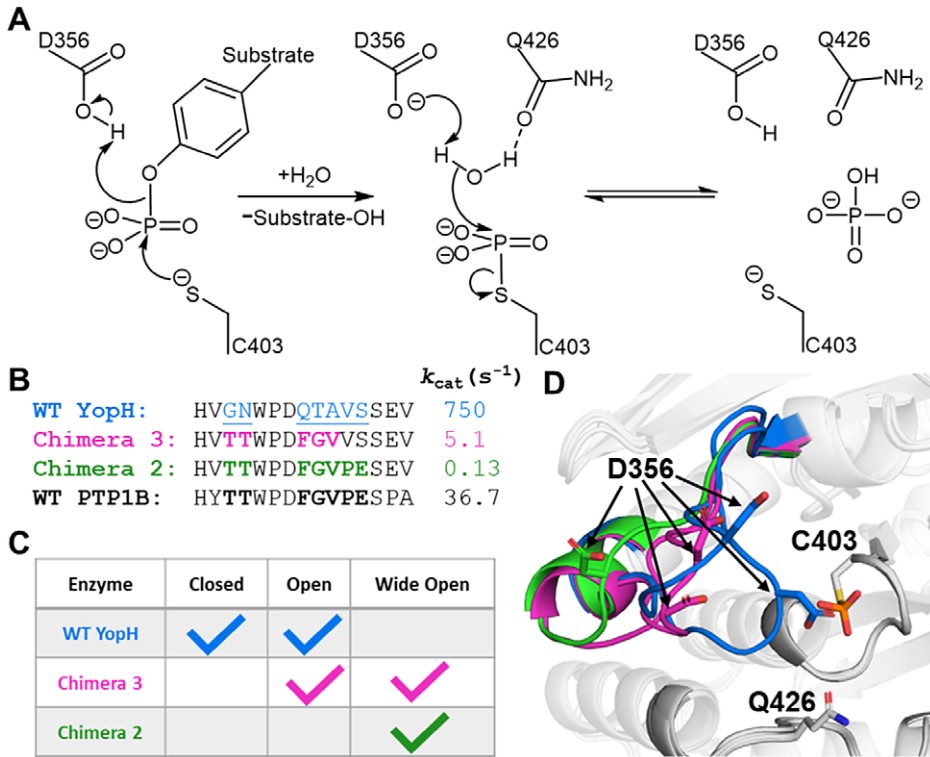

**Figure 1.** (*a*) Mechanism of PTP catalysis, with the residue numbering of wild-type (WT) YopH. (*b*) WPD-loop sequence alignments and corresponding turnover numbers of wild-type YopH, and YopH Chimeras 3 and 2, and wild-type PTP1B (from which the substitutions in the chimeras were obtained). Mutated chimera residues are shown in bold. Turnover numbers were measured at pH 5.5 and 25 °C using the substrate *p*NPP (Moise *et al.,* 2018. (*c*) Summary of the available crystal structures for each PTP in the different WPD-loop conformational states available. A tick indicates the availability of a crystal structure for a given PTP in a given conformational state. (*d*) A visualization of the diversity of WPD-loop conformational states captured by X-ray crystallography for wild-type YopH and Chimeras 2 and 3, and associated conformational diversity of the D356 side chain. Color coding of the structures WPD-loop matches that in panels (*b*) and (*c*). C403 is the nucleophilic cysteine, which is located on the P-loop, while Q426 is located on the Q-loop.

(Hedstrom *et al.,* 1992; James and Tawfik, 2003; Park *et al.,* 2006; Tawfik, 2006; Afriat-Jurnou *et al.,* 2012; Nestl and Hauer, 2014; Campbell *et al.,* 2016; Dodani *et al.,* 2016; Boehr *et al.,* 2018; Toogood and Scrutton, 2018; Dulcey *et al.,* 2019; Kundert and Kortemme, 2019; Crean *et al.,* 2020; Bunzel *et al.,* 2021; Damry and Jackson, 2021; Qu *et al.,* 2021; Ripka *et al.,* 2021; Schenk-mayerova *et al.,* 2021; Wiese *et al.,* 2021; Planas-Iglesias *et al.,* 2022). Furthermore, the use of chimeric proteins with grafted loops is a powerful tool both for gaining biochemical insight into specific enzyme systems and as an engineering tool (Hedstrom *et al.,* 1992; Park *et al.,* 2006; Tawfik, 2006; Doucet *et al.,* 2009; Clouthier *et al.,* 2012; Narayanan *et al.,* 2018; Dulcey *et al.,* 2019). In this context, loop-grafted chimeras of both PTP1B (Shen *et al.,* 2022) and YopH (Moise *et al.,* 2018), in which some or all of the WPD-loop of one enzyme is grafted onto the other enzyme's scaffold, show intriguing structural and biochemical properties. In the parent enzymes, the WPD-loop of PTP1B is more rigid than that of YopH, which has a highly conformationally flexible WPD-loop (Brandão *et al.,* 2012; Whittier *et al.,* 2013) and is also the most catalytic efficient PTP characterized to date (Zhang *et al.,* 1992).

Many of the chimeric offspring were inactive and/or insoluble, and those that were active showed reduced catalytic activities compared to that of either parent enzyme, despite conserved mechanisms and transition states (Moise *et al.,* 2018). Structural and computational characterization indicated that the catalytically active PTP1B chimeras maintained backbone structural integrity and were able to sample both open and closed conformations of the

WPD-loop, but with significant differences among them in the conformational space sampled (Shen *et al.,* 2022). The dynamics of the open WPD-loop are particularly complex, with the loop sampling multiple metastable and interconverting conformations. Furthermore, while empirical valence bond (EVB) simulations (Warshel and Weiss, 1980) indicated a chemical component to observed differences in turnover number, these were not substantial enough to account for the much larger observed effects (compared to differences in calculated activation free energies; Shen *et al.,* 2022).

In the case of YopH-based chimeras, transposing the WPD-loop of PTP1B onto the YopH scaffold resulted in two chimeric proteins with unusual structural properties, Chimeras 2 and 3 (Moise *et al.,* 2018). Crystal structures of both enzymes show their WPD-loop in a 'wide-open' conformation, facilitated by the extension of the adjacent α4 helix, which pulls the WPD-loop out into this conformation (Figure 1). While Chimera 2 is catalytically inactive, curiously, Chimera 3 retains some catalytic activity, albeit at a diminished rate compared to either parent enzyme (Supplementary Table S1). In addition, enhanced sampling (Hamiltonian replica exchange; Bussi, 2014) simulations indicate that although this wide-open structure has never been observed in any crystal structure of wild-type YopH, it *is* sampled even in the wild-type enzyme, although as a rare event (Crean *et al.,* 2021). Representative structures of YopH with wide-open WPD-loops show a similar α4-extension to that observed in the chimeras, suggesting that the chimeras did not generate this wide-open conformation, but rather that it is intrinsic also to

the wild-type sequence, and the chimeras merely stabilized it sufficiently to be observed in crystal structures.

This wide-open conformation is clearly catalytically inactive. However, Chimera 3, which crystallizes in this wide-open conformation in its unliganded form, shows catalytic activity comparable to an average PTP (Moise *et al.*, 2018, $k_{cat}$ 5.1 s$^{-1}$). Furthermore, similar wide-open conformations have been observed in three PTPs from different PTP subgroups: STEP, LYP, and GLEP1 (Barr *et al.*, 2009), pointing to a potentially functional role for this conformation, likely in the allosteric regulation of these enzymes. The putative importance of this conformation to allosteric regulation is further supported by the fact that small molecule inhibitors have been designed that displace the WPD-loop tryptophan of PTPRZ (Fujikawa *et al.*, 2017) and RPTPγ (Sheriff *et al.*, 2011), locking the WPD-loop into a similar wide-open conformation. More globally, it is intriguing that subtle perturbations to the WPD-loop sequence in both PTP1B and YopH can have such radical impact on the dynamical behavior of the loop, and thus also, in turn, the solubility and activity of the resulting constructs (Moise *et al.*, 2018; Shen *et al.*, 2022).

In the present work, we combine conventional and targeted molecular dynamics simulations with EVB simulations to characterize both the transition state for the rate-limiting hydrolysis step catalyzed by Chimeras 2 and 3, as well as the dynamical properties of the WPD-loops of both YopH chimeras. We observe negligible energetic differences between the chemical steps of the different enzymes, but significant differences between their WPD-loop dynamics, and provide insight into the detailed molecular interactions driving these differences. Taken together, our results further emphasize both the power of loop engineering as a simple strategy to manipulate enzyme physiochemical properties, as well as the challenges involved due to the unpredictability of the behavior of the resulting constructs.

## Methods

### EVB simulations

We have used EVB simulations (Warshel and Weiss, 1980) to model the rate-limiting hydrolysis step catalyzed by Chimeras 2 and 3, following from our prior simulation studies of PTP1B and YopH (Crean *et al.*, 2021; Shen *et al.*, 2022). We applied the same simulation setup, including equilibration and EVB simulations, as in our prior work (Crean *et al.*, 2021; Shen *et al.*, 2022), using the revised parameters provided in Crean *et al.* (2022). EVB simulations were performed using the *Q6* simulation package (Bauer *et al.*, 2018) and the OPLS-AA (Jorgensen *et al.*, 1996) force field, for consistency with our previous work (Crean *et al.*, 2021; Shen *et al.*, 2022). Each chimera was simulated for 30 replicas, using an initial 30 ns of equilibration starting from the approximate EVB transition state (λ = 0.5, Supplementary Figure S1). Production simulations were then propagated from the transition state to both reactants and products using 51 mapping windows in total, each with a simulation time of 200 ps. Simulation analysis was performed using CPPTRAJ (Roe and Cheatham, 2013).

### System preparation for MD simulations

Molecular dynamics (MD) simulations of wild-type YopH and Chimeras 2 and 3 were performed in the phospho-enzyme intermediate state (the starting state for the second rate-limiting hydrolysis step). We have previously (Crean *et al.*, 2021) performed MD simulations on the closed and open WPD-loop states of wild-type YopH, and, in this work, we used these prepared systems to simulate wild-type YopH (and prepared the structures of Chimeras 2 and 3 in a consistent manner). Briefly, the Amber ff14SB force field (Maier *et al.*, 2015) and TIP3P water model (Jorgensen *et al.*, 1983) were used to describe protein and water molecules, with simulations performed using Amber 18 (Case *et al.*, 2018). Simulations were performed at constant temperature and pressure (298 K, 1 atm), with a 2 fs time step. All systems were equilibrated for production MD simulations using the same protocol, which is described in full in the Supplementary Material. Each production simulation was 500 ns in length, and performed in 15 replicates. The convergence of these simulations is shown in Supplementary Figures S2–S4.

### Targeted molecular dynamics simulations

Targeted molecular dynamics (tMD; Schlitter *et al.*, 1994) simulations were performed using Amber 18 (Case *et al.*, 2018) interfaced with PLUMED v2.5 (Tribello *et al.*, 2014). Four systems were subjected to tMD simulations, and 20 replicas were generated per system. System preparation was performed as described in the Supplementary Material. The proteins were steered toward the closed WPD-loop conformation by using the backbone RMSD to the crystal structure conformation of WT-YopH in the closed conformation (PDB ID: 2I42; Denu *et al.*, 1996) as the collective variable/reaction coordinate. After testing several combinations of steering forces and pulling times, we settled on using a pulling force of 75 kcal mol$^{-1}$ over the course of 30 ns as this combination satisfied Jarzynski's equality (RMSD vs. time is a relatively smooth diagonal line, see Figure 3 and Supplementary Figure S5, while being relatively fast). After first equilibrating each structure to a starting RMSD of 5.0 Å over the course of 5 ns, the backbone RMSD to the target was then progressively decreased to be at 1.0 Å at the end of the 30 ns. Following this, the steering force was progressively removed over the course of 5 ns and a further 10 ns of unrestrained (effectively normal MD simulations) were performed.

## Results and discussion

### EVB simulations of Chimeras 2 and 3

As our starting point, we performed EVB simulations of Chimeras 2 and 3 for comparison to our prior simulations of wild-type PTP1B and YopH (Crean *et al.*, 2021, 2022). As in prior work (Shen *et al.*, 2022), we focused here on modeling just the second hydrolysis step (Figure 1), as this is rate-limiting (Keng *et al.*, 1999; Cui *et al.*, 2019). Important to note is that due to the lack of ligand-bound crystal structures of either chimera with the WPD-loop in its catalytically active closed conformation, simulations were initiated from the wild-type YopH crystal structure (PDB ID: 2I42; Berman *et al.*, 2000), with the relevant WPD-loop substitutions for each chimera modeled in manually to mimic a hypothetical closed structure of each chimera, as described in the Methods section. This thus represents an 'idealized' conformation of the active site and WPD-loop which is not necessarily sampled in reality; however, starting from this idealized conformation is useful in that it allows us to distinguish between chemical effects caused by the loop substitutions (that would manifest even in an idealized loop-closed conformation) and those caused by loop dynamics (including an inability to sample the narrowly defined (Crean *et al.*, 2021 closed conformation). Specifically, we know that Chimera 2 and Chimera

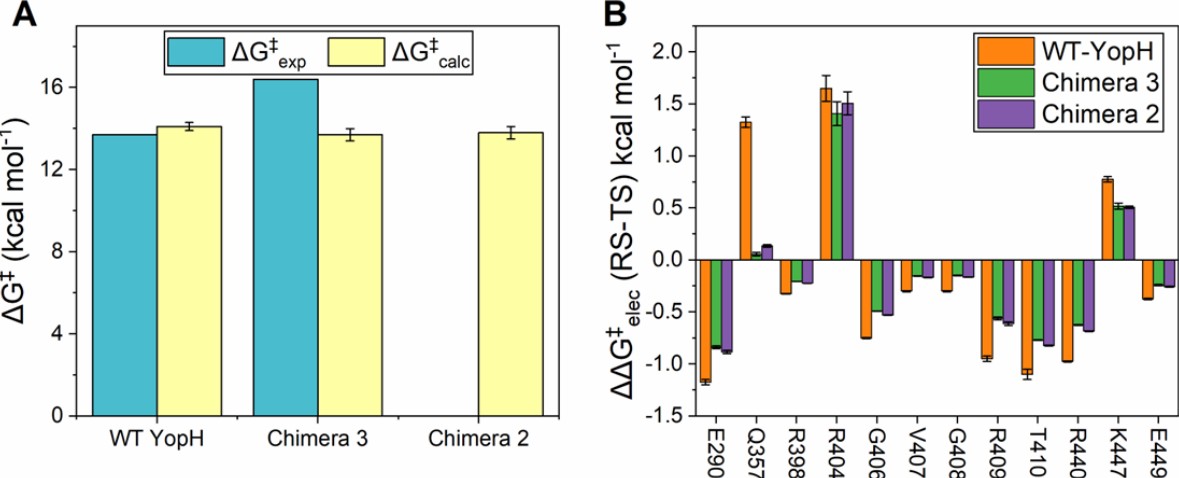

**Figure 2.** (*a*) Comparison of the calculated ($\Delta G^{\ddagger}_{calc}$) and experimental ($\Delta G^{\ddagger}_{exp}$) activation free energies calculated for the hydrolysis step of the reaction catalyzed by wild-type (WT) YopH and Chimeras 3 and 2. Simulation data are presented in kcal mol$^{-1}$ as the average values and standard error of the mean obtained from 30 EVB simulation replicas. Experimental data are obtained from (Zhang *et al.,* 1992; Stuckey *et al.,* 1994; Whittier *et al.,* 2013; Moise *et al.,* 2018) The raw data for this figure are presented in Supplementary Table S1. Note that no experimental data are presented in panel A for Chimera 2, as this Chimera is catalytically inactive (Moise *et al.,* 2018). (*b*) Electrostatic contributions of selected amino acids ($\Delta\Delta G^{\ddagger}_{elec}$, kcal mol$^{-1}$) to the calculated activation free energies for the hydrolysis steps catalyzed by WT-YopH and Chimeras 2 and 3. All electrostatic contributions were scaled assuming an internal dielectric constant of 4 (Li *et al.,* 2013). Data were obtained from the calculated EVB trajectories using the linear response approximation (LRA)(Lee *et al.,* 1992; Muegge *et al.,* 1997) and are represented as average and standard error of the mean over 30 individual trajectories per system. The amino acids that directly participate in the reaction (the catalytic cysteine and aspartic acid) are not shown. Chimeras 2 and 3 carry a Q357F substitution. The corresponding raw data for this plot are presented in Supplementary Table S3.

3 both sample wide-open conformations, at least in their unliganded forms, and that Chimera 3 retains activity (albeit diminished compared to wild-type), while Chimera 2 is inactive (Moise *et al.,* 2018). Our EVB simulations aim to address whether there are differences at the level of the active sites of the closed conformation of each chimera that led to the loss of activity in these constructs, or whether the loss of activity is more likely to be a result of the inability of the WPD-loop to sample a catalytically optimal closed conformation, in particular in Chimera 2, which is not catalytically active. We note that our prior work has suggested a strong link between WPD-loop sequence and the conformational dynamics of the WPD-loop in chimeras with some or all of the YopH WPD-loop grafted onto the PTP1B scaffold (Shen *et al.,* 2022). Furthermore, the PTP1B chimeras shifted to a closed conformation of the WPD-loop, in contrast to the YopH chimeras studied here.

The calculated activation and reaction free energies from our EVB calculations of the hydrolysis step catalyzed by wild-type YopH and PTP1B (prior work; Crean *et al.,* 2021, 2022) as well as YopH Chimeras 2 and 3 are shown in Supplementary Table S1 and Figure 2, with structures of representative stationary points from our simulations shown in Supplementary Figure S6, with the corresponding average reacting distances for each relevant variant summarized in Supplementary Table S2. As can be seen from this data, if anything, we obtain slightly *lower* calculated activation free energies for Chimeras 2 and 3 compared to what we obtain for the wild-type parent enzymes. This is similar to our prior work on PTP1B chimeras, where the small differences in predicted activation free energies from idealized closed conformations were inadequate to account for much larger differences in turnover numbers, in contrast to the much larger differences in loop dynamics between chimeras observed in our simulations (Shen *et al.,* 2022).

Related to this, transition state geometries and solvent penetration of the active site (Supplementary Table S2), as well as electrostatic contributions of individual amino acids to catalysis

(Figure 2*b* and Supplementary Table S3), are virtually unchanged between wild-type YopH and the chimeric constructs (the largest ~1 kcal mol$^{-1}$ difference comes from residue 357, which is a Gln in wild-type YopH, and a Phe in Chimeras 2, 3, and the corresponding position in wild-type PTP1B). This is perhaps unsurprising because no active site residues were perturbed upon creation of these constructs, and our simulations emphasize that, were the Chimeras to be able to achieve optimal closed conformations of the WPD-loop, they would be expected to show similar turnover numbers to the parent enzymes. Thus, the diminished activity is highly likely to be due to alterations in WPD-loop dynamics and not chemistry upon creation of these chimeric proteins.

### Targeted MD simulations

Given our EVB results implied that the catalytic differences observed between the Chimeras and wild-type YopH were linked to the conformational sampling of the WPD-loop, we decided to explore the conformational space available to each PTP and compare the relative stabilities of the different conformational states. As there are no crystal structures available of Chimeras 2 and 3 with a closed WPD-loop conformation, we used two strategies in order to generate structures of the Chimeras with a closed WPD-loop conformation. The first strategy was to mutate (*in silico*) the wild-type YopH closed loop structure to become Chimeras 2 and 3 and perform MD simulations starting from this conformation, which will be described later. The second strategy employed was to perform targeted MD (tMD) simulations (Schlitter *et al.,* 1994) starting from the 'wide-open' WPD-loops from both conformations. Using tMD, we slowly 'steered' the WPD-loop of a given Chimera from the 'wide-open' to the 'closed' WPD-loop conformation over the course of an MD simulation. To do this, we used the wild-type YopH closed WPD-loop crystal

structure as a reference structure, and applied a restraint to the backbone RMSD atoms of the WPD-loop to enforce a slow transition between the two states. The results of our tMD simulations for both Chimeras are presented in Figure 3, and for wild-type YopH (as a control) are presented in Supplementary Figure S5. For all systems, we generated 20 replicas per system, pulling the conformations to the closed WPD-loop over the course of 30 ns. Following this, the pulling force was progressively removed over the course of 5 ns, and 10 ns of unrestrained MD simulations were then performed. Analysis of Figure 3a,b shows a large spread of RMSDs across the replicas once the restraints were completely released (the last 10 ns of each replica), meaning that in some replicas the closed WPD-loop conformation was retained, while in others the WPD-loop re-opened rapidly. Similar observations were also made in the control simulations of wild-type YopH starting from a wide-open conformation (Supplementary Figure S5A,B).

To identify potential causes for why some tMD simulations gave rise to relatively stable closed WPD-loop conformations while others did not, we investigated the last 5 ns of each replica. At this time point, no restraints would have been applied to the system for at least 5 ns, meaning the structure would have had time to somewhat 'relax'. Our analysis revealed the requirement for the P-loop residue R409 to be coordinating the phosphate group in order to facilitate proper closure of the WPD-loop (Figure 3d). That is, R409 can adopt two major conformations as depicted in Figure 3d, in which the non-coordinated conformation sterically blocks productive loop closure. By comparing the average RMSD for the last 5 ns of each replica to the R409 phosphate group distance, we observe a clear requirement that this conformation has to be adopted in order for productive loop closure to be possible. It is important to note that the correct conformation of R409 did not guarantee productive loop closure, in all instances, however. This observation helps to explain the population shift toward the closed state that is known to occur once a phosphate group is bound (covalently or non-covalently) to the active site. That is, the phosphate groups help to place the R409 side chain in the correct position for productive loop closure,

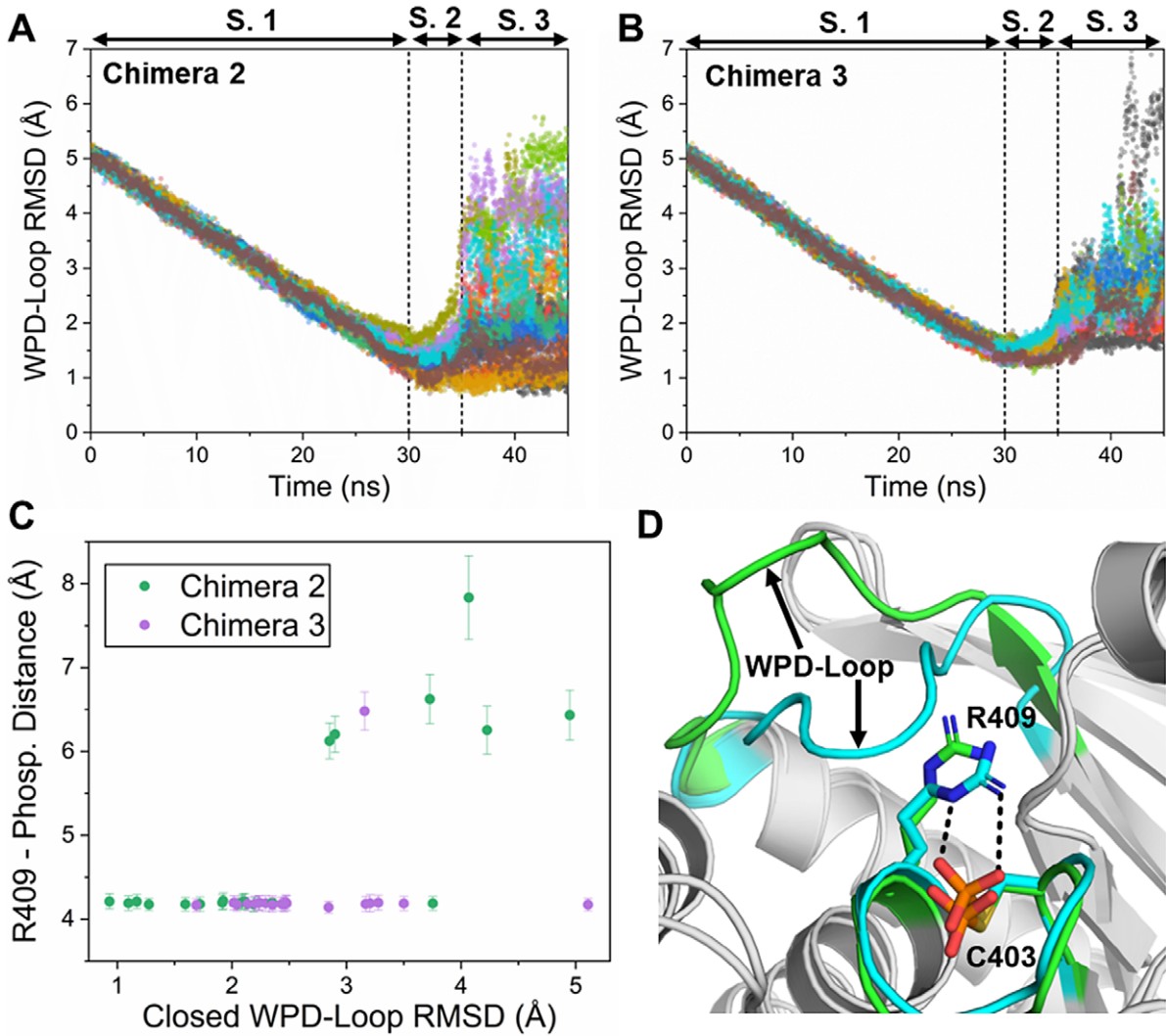

**Figure 3.** (a,b) The RMSD to the closed WPD-loop conformation over the course of our targeted MD (tMD) simulations of YopH Chimera 2 (a) and Chimera 3 (b). The reference state used is the backbone atoms of the closed WPD-loop structure of YopH. (c) Center of mass distance between the P-loop R409 side chain and the phosphorylated cysteine against the RMSD to the closed WPD-loop for the last 5 ns of each tMD simulation for both Chimeras 2 and 3. Errors are the standard deviation obtained from each frame. (d) Representative structures of the productive and non-productive R409 side-chain conformations, with the productive (for WPD-loop closure) structure colored in cyan and the non-productive structure (the R409 would sterically clash with the WPD-loop if it was closed) colored in green.

as also observed in our earlier study of wild-type PTP1B and YopH (Crean *et al.,* 2021).

### Evaluating the differences in stability of the WPD-loop conformations

Our tMD simulations identified that it is indeed possible to form 'closed' WPD-loop conformations for both Chimeras, and that this conformation can be stable (at least on short simulation timescales) in these PTP variants. Likewise, wild-type YopH is known to be able to form a closed WPD-loop conformation, as it has been crystalized in that conformation (Berman *et al.,* 2000), unlike the two Chimeras (Moise *et al.,* 2018). Taken together, this would suggest all enzymes can sample the three major WPD-loop conformational states (closed, open, and wide-open) but with altered favorability among the states. We wished to investigate this proposed

population shift across the enzymes and used MD simulations starting from each conformational state to do so. That is, each enzyme was simulated using 15 replicas of 500 ns long MD simulations starting from each conformational state. We note that initial attempts to build Markov State Models (Chodera and Noé, 2014) of the WPD-loop state using these simulations were unsuccessful due most likely to insufficient sampling between the closed, open, and wide-open WPD-loop states as depicted in Supplementary Figure S7. We instead therefore evaluated the relative stability of the WPD-loop for each system in each conformational state by measuring the $C_\alpha$-atom RMSD of the WPD-loop residues throughout each simulation and used this to generate probability distributions presented in Figure 4a–c. In cases where crystal structures of the conformational state to be simulated were not available for a given enzyme, the required residues were substituted *in silico*, see the Methods for further details.

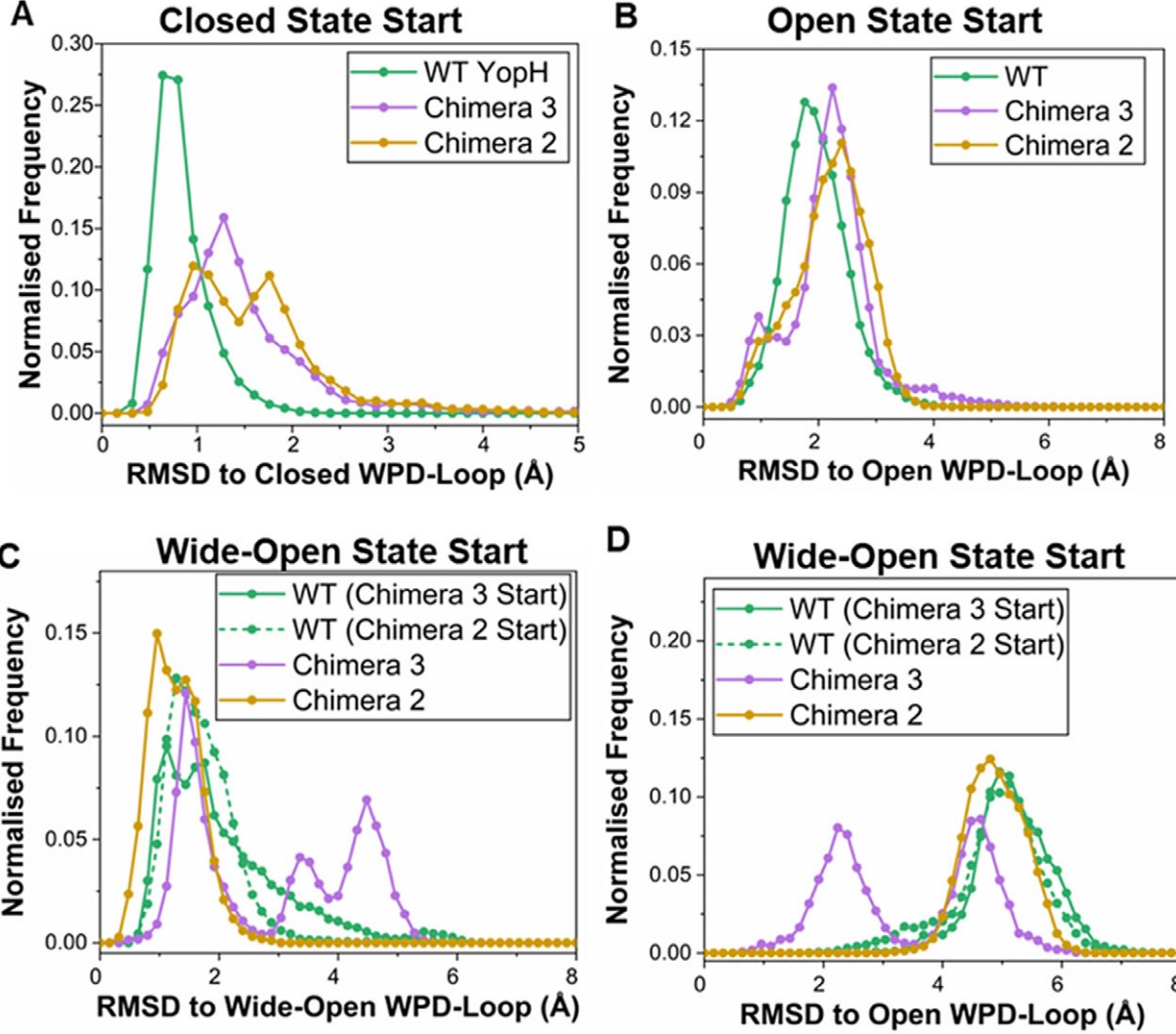

**Figure 4.** Evaluating the stability of the closed, open, and wide-open WPD-loop conformational states in simulations of wild-type (WT) YopH and Chimeras 2 and 3. (*a*) Histograms of the WPD-loop $C_\alpha$-atom RMSD to the closed WPD-loop conformational state, with simulations starting from the closed conformational state. The closed conformation Chimera structures were generated by introducing the relevant substitutions into the wild-type YopH WPD-loop *in silico*, as described in the Methods section. (*b*) Histograms of the WPD-loop $C_\alpha$-atom RMSD to the open WPD-loop conformational state, with simulations starting from the open conformational state. (*c*) Histogram of the WPD-loop $C_\alpha$-atom RMSD to the wide-open conformational state for simulations initiated from the wide-open conformational state of the WPD-loop. (*d*) Histogram of the WPD-loop $C_\alpha$-atom RMSD to the open conformational state for simulations initiated from the wide-open conformational state of the WPD-loop. For panels (*c*) and (*d*) which refer to the wide-open simulations, the two wild-type YopH wide-open loop conformations were constructed by introducing the relevant substitutions into both the Chimera 3 and Chimera 2 WPD-loop residues *in silico*, see the Methods section. These are indicated on the graph legend as 'WT (Chimera 3 Start)' if produced using the Chimera 3 crystal structure or 'WT (Chimera 2 Start)' if produced using the Chimera 2 crystal structure. In all cases, each histogram was constructed using 15 × 500 ns long MD simulation replicas, sampling data every 100 ps, with a bin width of 0.16 Å.

Focusing first on the closed WPD-loop simulations (Figure 4*a*), the simulations show the wild-type YopH WPD-loop closed conformation to be the most stable conformation, which samples a narrower and smaller distribution of RMSD values compared to Chimeras 3 and 2. For example, 95.7% of all wild-type YopH frames have an RMSD ≤1.5 Å from the crystallographic closed conformation, as compared to 57.4% and 45.8% of frames from simulations of Chimeras 3 and 2, respectively. This measurement is robust to different RMSD cutoffs, for example, the percentage of frames that have an RMSD ≤2.0 Å for each enzyme are 99.6%, 81.6%, and 75.8% for the wild-type enzyme, Chimera 3, and Chimera 2, respectively. The observation that wild-type YopH has a notably more stable closed WPD-loop conformation is consistent with the aforementioned crystallographic data (Berman *et al.,* 2000; Moise *et al.,* 2018). Further, visual inspection of the individual RMSD traces used to build the histograms depicted in Figure 4*a* identified clear examples of WPD-loop opening in Chimeras 2 and 3 (Supplementary Figure S8). This is in contrast to the simulations of wild-type YopH, where WPD-loop opening is notably more rarely observed (Supplementary Figure S8). Given the similarity in the probability distributions and the number of observations, it is unclear if there is a difference in the relative stability of the closed WPD-loop state for Chimeras 3 and 2.

Our simulations of the open conformational state (Figure 4*b*) showed no obvious differences between the three different enzymes, suggesting that the relative conformational stability of this state was not notably perturbed as a result of the substitutions that separate each enzyme. However, our simulations of the wide-open state (Figure 4*c*) did however show clear differences in the relative stabilities of the different enzymes. We note that while we have observed the wide-open conformation of the WPD-loop as a rare event in simulations of wild-type YopH (Crean *et al.,* 2021), there is no available crystal structure of the wild-type enzyme in this conformation. We therefore generated independent starting structures for our wide-open wild-type YopH simulations *in silico*, using both the Chimera 3 and 2 structures as models for this conformation of the WPD-loop. Reassuringly, these two different wide-open starting structures of the wild-type YopH WPD-loop sampled similar distributions, and were observed to be approximately equally stable, independent of starting conformation (Figure 4*c*). A clear difference in the stability of the wide-open conformational state was observed between Chimeras 3 and 2 (Figure 4*c*), which differ by two substitutions between them (V360P and S361E). That is, the wide-open state of Chimera 2 is notably more stable over the course of our simulations than that of Chimera 3. Simulations of Chimera 3 starting from the wide-open conformation show that in several replicas, the WPD-loop leaves the wide-open conformation (as evidenced by the distribution of RMSD values ≥2.5 Å in Figure 4*c*). The RMSD of these simulations to the open WPD-loop conformation shows that these replicas have transitioned from the wide-open to the open conformation (Figure 4*d*).

By combining our RMSD analysis presented above with the available crystallographic data (Moise *et al.,* 2018), we can propose the following population shifts between wild-type YopH and Chimeras 3 and 2. First, the five substitutions that separate wild-type YopH and Chimera 3 are responsible for destabilizing the closed WPD-loop conformational state, with similar instability observed for Chimeras 3 and 2 (which are separated by a further two substitutions). Second, the two substitutions that separate Chimeras 3 and 2 are responsible for the observed increased stability in the wide-open conformation of Chimera 2. This increased

propensity to populate the catalytically inactive wide-open conformation will clearly contribute to the impaired catalytic activity of this variant, both compared to wild-type YopH, and compared to Chimera 3 (Moise *et al.,* 2018).

### Molecular basis for the observed population shifts

Our MD simulations (Figure 4) identified two major population shifts as a result of the WPD-loop substitutions: one in the closed state as a result of the mutations that separate wild-type YopH and Chimera 3 (as Chimeras 3 and 2 behave similarly in this state); and one in the wide-open state as a result of the two mutations that separate Chimeras 3 and 2 (as wild-type YopH and Chimera 3 behave similarly in this state). Having observed *that* there is a significant population shift, the next important question is *how* the WPD-loop substitutions which separate these enzymes gave rise to these observed population shifts. Focusing first on the differences between wild-type YopH and Chimera 3, we determined the average difference in the $C_\alpha$-root-mean-squared fluctuations (ΔRMSFs) for every residue in each enzyme from our simulations of the closed WPD-loop conformation (Supplementary Figure S9A). To evaluate if each residue's calculated ΔRMSF was significant, we performed a two-sided *t*-test using the 15 replicas performed for each enzyme as input.

Further, to account for the usage of many *t*-tests (282 *t*-tests, 1 for each residue), we applied the Benjamini–Hochberg correction (Benjamini and Hochberg, 1995), using a false discovery rate of 5%. This effectively means 5% of the features identified as significant will be false positives, whereas without the correction, this number would be notably higher. We note that the regions with the largest ΔRMSF values (~residues 220–230) in Supplementary Figure S9A were not determined to be significantly different. This makes sense due to this region of the protein being highly flexible, and sampling large RMSF values in both enzymes (Supplementary Figure S10). This also showcases the importance in applying some form of statistical testing prior to interpreting MD simulation results such as these. Supplementary Figure S9A,B identifies three major regions (the WPD-loop, the β5- and β6- strands, and the α7-helix) where there is a significant difference in RMSF values between wild-type YopH and Chimera 3. Consistent with the RMSD data, the closed WPD-loop conformation is more stable in wild-type YopH as compared to Chimera 3, especially at the central portion of the WPD-loop. To understand why these regions are more stable in wild-type YopH, we determined how the non-covalent interaction networks differed in both proteins using our recently developed package, Key Interactions Finder (KIF, Crean *et al.,* 2023). In order to perform KIF analysis, we labeled each simulation frame as belonging to one of the following four states: 'closed', 'open', 'wide-open', and 'other', with simulation frames belonging to 'other' discarded from the analysis. The approach used to label each frame is described in the Supplementary Methods.

The results obtained from KIF are presented in Supplementary Figures S9C,D and S11. By analyzing these differences, we identified the T358G substitution resulted in the removal of two inter-WPD-loop hydrogen bonding interactions with W354 and P355. This helps to explain the likely decrease in stability of the central portion of the WPD-loop, as a result of this substitution.

Our interaction network data also show an increased number of interactions for Chimera 3 at the N-terminal portion of the WPD-loop (Supplementary Figure S9C,D) as a result of the G352T and N353T substitutions. Consistent with this, our RMSF calculations determined this part of the WPD-loop to be more rigid in Chimera

3 than in wild-type YopH, but failed to reject the null hypothesis (Supplementary Figure S9A,B). That said, given that the *p*-values associated with these two mutated residues are 0.017 and 0.026 respectively, it is likely this region of the protein is more stable in Chimera 3 than in wild-type YopH (note that the use of Benjamini–Hochberg correction (Benjamini and Hochberg, 1995) for multiple statistical tests means that *p*-values <0.05 are not automatically considered significant, see the Supplementary Methods section for further details).

Two substitutions on the WPD-loop (V360P and S361E, see Figure 1) separate Chimeras 2 and 3, and therefore give rise to the observed increase in stability of the wide-open state for Chimera 2, as seen in our MD simulations. To probe this further, we determined the ΔRMSF and the differences in the interaction networks (using KIF) for Chimeras 2 and 3 when sampling the wide-open conformational state (Figure 5), in order to identify how these substitutions contributed to this population shift. Our RMSF calculations revealed the central portion of the WPD-loop Chimera 2 structure to be substantially more rigid than in Chimera

3 (Figure 5*a,b*). Further, the α-helix which follows the WPD-loop (which contains the two substitutions V360P and S361E) was also observed to be significantly more stable in Chimera 2 (Figure 5*a,b*). Interestingly, the N-terminal portion of the WPD-loop was observed to be significantly more stable in Chimera 3, although the magnitude of these changes was comparatively lower (~3 Å vs. 0.5 Å ΔRMSF differences, see Figure 5*a*).

Our KIF calculations (Supplementary Figure S13) identified the largest change in the interaction network to be the removal of an inter-loop hydrogen bond between the backbone of V360P to F357 (Figure 5*c,d*). This interaction is present in Chimera 3 but abolished in Chimera 2 due to the proline nitrogen no longer being able to form a hydrogen bond (a proline does not have a hydrogen on the nitrogen group). Given the nature of the substitution (to a conformationally constricted residue), we also analyzed the backbone dihedral angles sampled by V360P in both the wide-open and open states of Chimeras 2 and 3 and observed no sampling of disallowed Ramachandran values (Supplementary Figure S14). This would suggest that the observed population shift was not a result of any

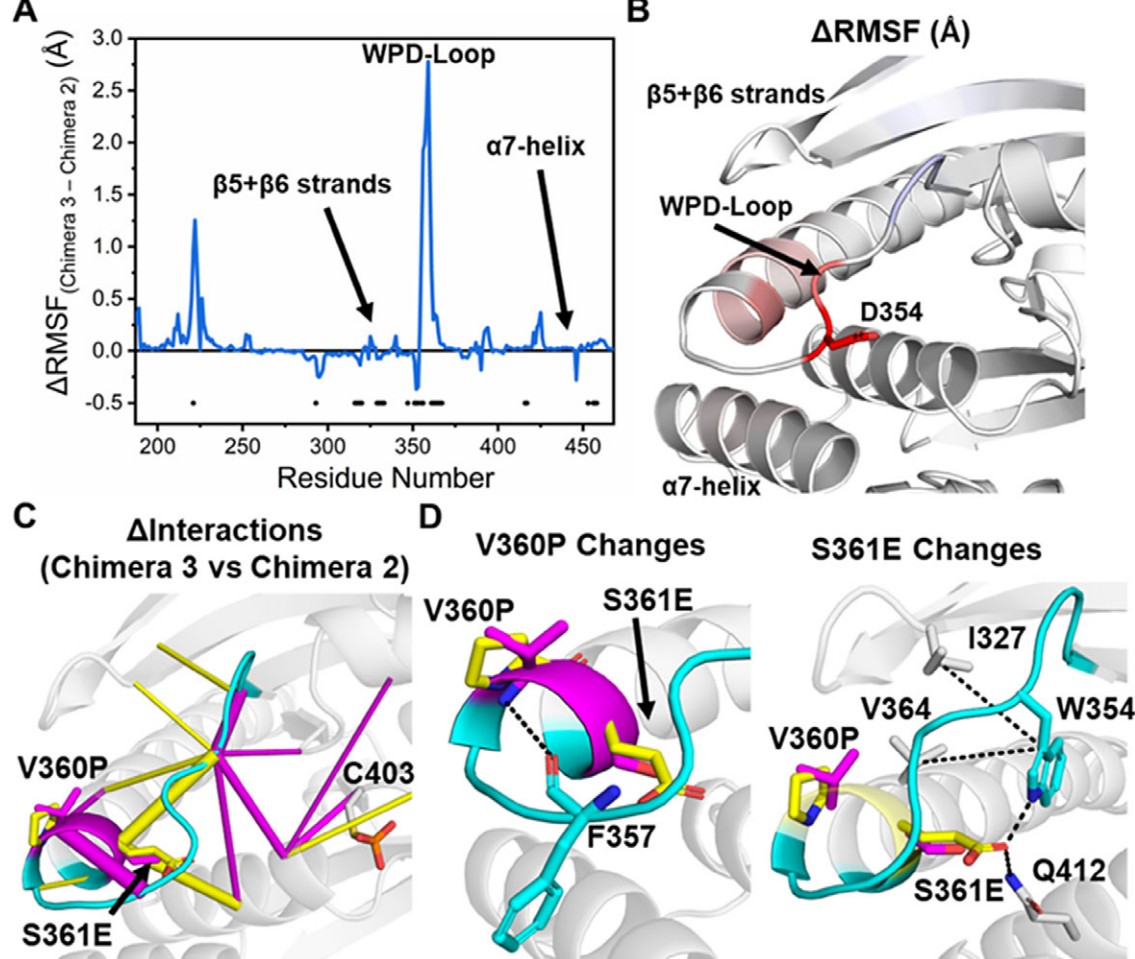

**Figure 5.** (*a*) Difference in the calculated root-mean-squared fluctuations (ΔRMSF) of Chimeras 3 and 2 when sampling the wide-open WPD-loop conformational state. A two-sided *t*-test was performed to validate the significance of the calculated ΔRMSFs, and those residues identified as significant have a black dot placed at the bottom of the graph. To account for the usage of multiple *t*-tests, the Benjamini–Hochberg correction (Benjamini and Hochberg, 1995) was applied, using a false discovery rate of 5%, see the Methods section for further details. (*b*) Projection of the calculated ΔRMSFs identified as significantly different onto the structure of YopH, with residues colored from blue (more rigid in the Chimera 3), to white (equally rigid or no significant difference) to red (more rigid in Chimera 2). RMSF profiles of both enzymes are provided in Supplementary Figure S12. (*c*) Differences in the non-covalent interaction network between Chimera 3 and Chimera 2 when sampling the wide-open WPD-loop conformational state as determined by KIF (Crean *et al.*, 2023). Interactions which are on average stronger in the Chimera 3 are colored magenta, while those that favor Chimera 2 are colored yellow. These data are shown graphically in Supplementary Figure S12. (*d*) Illustration of the major changes in interactions at the two substitution sites (V360P and S361E), which were depicted in panel (*c*).

steric hindrance as a result of the addition of a proline residue. Taken together, the V360P substitution would not be expected to give rise to the observed population shift, as the only change in the interaction network would be expected to *increase* the stability of the wide-open state for Chimera 3.

In contrast to the V360P substitution, the increased stability of the wide-open state observed for Chimera 2 can be rationalized by the S361E substitution, which enables Chimera 2 to form an intra-loop hydrogen bond with W354 and an inter-loop hydrogen bond with Q412 (Figure 5d). This is consistent with our RMSF calculation in that the central portion of the WPD-loop becomes notably more stable in Chimera 2 over Chimera 3 (Figure 5a,b).

## Conclusions

Recent years have seen increasing awareness of the evolutionary role of active site loop dynamics (Pinto *et al.,* 2021), as well as the implications of this for enzyme engineering (Nestl and Hauer, 2014; Pinto *et al.,* 2021). PTPs are an excellent model system to probe this, as there is both computational and experimental evidence (Whittier *et al.,* 2013; Crean *et al.,* 2021; Shen *et al.,* 2021, 2022) suggesting a direct link between the dynamics of the active-site WPD-loop and the turnover rate of these enzymes. Furthermore, these enzymes are biomedically important but extremely challenging drug targets, in particular for the development of novel cancer therapeutics, and the design of allosteric inhibitors that can impair proper closure of the WPD-loop remains a promising avenue for drug discovery efforts targeting these enzymes (Wiesmann *et al.,* 2004; Sheriff *et al.,* 2011; Fujikawa *et al.,* 2017; Keedy *et al.,* 2018).

The WPD-loops of PTPs are conformationally flexible, and can take on a range of open and even wide-open conformational states (Crean *et al.,* 2021; Shen *et al.,* 2022). Further, measurement of kinetic isotope effects for the reactions catalyzed by both wild-type PTP1B and YopH (Hengge *et al.,* 1995; Brandão *et al.,* 2012) has shown that the leaving group is effectively neutralized in the transition state by protonation. This provides a strong constraint on loop pose, in that the loop must be sufficiently closed in order for the catalytic Asp (on the WPD-loop) to be in range for proton transfer. The KIEs between wild-type YopH and the Chimeras studied herein differ slightly (Moise *et al.,* 2018), but are still within range of those observed for other PTPs (Hengge *et al.,* 1995; Brandão *et al.,* 2012; Hengge, 2015; Moise *et al.,* 2018). That is, the proton transfer is sufficiently coordinated with P-O bond fission that the leaving group remains neutral, meaning that despite this conformational plasticity, the 'catalytically productive' conformations available to the enzyme are limited. This observation also aligns will with our prior computational work comparing the impact of loop conformation against the chemical barrier for the enzyme triosephosphate isomerase (TPI; Liao *et al.,* 2018). While there is no catalytic residue on the TPI catalytic loop, our simulations demonstrated the need for a near perfect closed conformation (compared to crystallographic data; Jogl *et al.,* 2003) in order for the activation barrier to not increase significantly. The fact that in the present case there is additionally also a key catalytic residue directly involved in proton transfer located on the mobile loop, it is extremely unlikely for chemistry to be viable from a partially closed WPD-loop conformation, and therefore the ability to reach a productive closed conformation is crucial for catalysis.

Molecular dynamics simulations of a range of PTP1B chimeras, in which some or all of the YopH WPD-loop was grafted onto the PTP1B scaffold, have indicated that there are significant differences in the conformational dynamics of the WPD-loop both compared to either parent enzyme, and when comparing the dynamics of the individual chimeras (Shen *et al.,* 2022). Following from this, both computational and crystallographic analysis indicated the substitutions cause a population shift toward a WPD-loop closed conformation (Shen *et al.,* 2022). This is the inverse of structural characterization of YopH chimeras in which parts of the PTP1B WPD-loop was grafted onto the YopH scaffold, and where substituting amino acids from the PTP1B WPD-loop into the YopH loop triggered instead a wide-open conformation in the chimeric proteins (Chimeras 2 and 3; Moise *et al.,* 2018). Of these chimeras, only Chimera 3 was shown to have meaningful catalytic activity, although similar to the PTP1B chimeras (Shen *et al.,* 2022), the activity of this chimeric construct was impaired compared to that of either parent enzyme (Moise *et al.,* 2018).

The wide-open conformation observed in these YopH chimeras has previously been observed in other PTPs from different subgroups, STEP, LYP, and GLEP1 (Barr *et al.,* 2009), as well as being sampled as a rare event in simulations of wild-type YopH (Crean *et al.,* 2021). Further, small molecule inhibitors have successfully triggered this wide-open conformation in PTPRZ (Fujikawa *et al.,* 2017) and RPTPγ (Sheriff *et al.,* 2011), indicating the importance of understanding the relevance of this conformation for drug discovery efforts.

Here, we have simulated both Chimeras 2 and 3 from (Moise *et al.,* 2018) as well as wild-type YopH, starting from different conformations of the WPD-loop, and combining conventional and targeted molecular dynamics simulations with EVB simulations of the rate-limiting hydrolysis step. As in our prior work (Crean *et al.,* 2021), EVB simulations initiated from idealized closed conformations suggest negligible energetic differences between the different enzyme variants, which is to be expected as these enzymes have essentially identical active site architectures that are not modified by the alterations in WPD-loop sequence. This points toward a putative dynamical origin for the differences in activity. Our conventional and targeted molecular dynamics simulations indicate significant differences in the stability of the wide-open conformation of the WPD-loop, which is more stable in inactive Chimera 2 than in active Chimera 3 or wild-type YopH, suggesting that it is more challenging for Chimera 2 to sample a catalytically active conformation.

More detailed analysis of the molecular basis for the differences in stability of the loop indicated that the central portion of the WPD-loop of Chimera 2 is substantially more rigid than that of Chimera 3, and the α-helix at the C-terminus of the WPD-loop is significantly more stable in Chimera 2 (Figure 5). We note that our prior computational study of wild-type YopH indicated the importance of this α-helix to being able to populate a wide-open conformation of the loop in the first instance (Crean *et al.,* 2021). Our simulations also provide insight into subtle differences in the interaction networks controlling loop motion between the different enzymes, and its link to loop stability and activity. Taken together, our data provide insight into the regulation of the wide-open conformation of YopH and the implications this has for catalysis, which is an important steppingstone for both understanding the functional relevance of this conformation, and how it can be exploited for drug discovery purposes.

**Open peer review.** To view the open peer review materials for this article, please visit http://doi.org/10.1017/qrd.2024.3.

**Supplementary material.** The supplementary material for this article can be found at https://doi.org/10.1017/qrd.2024.3.

**Acknowledgments.** We would like to thank Dr. Bruno di Geronimo Quintero for bringing to our attention the wide-open conformation observed in PTPRZ and RPTPγ.

**Author contribution.** R.M.C., A.C.H., and S.C.L.K. conceived and designed the study. R.M.C., M.C., and A.R.C. performed calculations. All authors analyzed data. R.M.C. and S.C.L.K. wrote the original draft of the manuscript. All authors contributed to revising and editing the manuscript. S.C.L.K. secured funding and computational resources.

**Financial support.** This work was supported by the Knut and Alice Wallenberg Foundation (grant numbers 2018.0140 and 2019.0431), the Swedish Research Council (grant number 2019-03499), and the Carl Tryggers Foundation for Scientific Research (grant number CTS 19:172). The project has received funding from the European Union's Horizon 2020 research and innovation program under the Marie Skłodowska-Curie grant agreement No. 890562. The simulations were enabled by resources provided by the Swedish National Infrastructure for Computing (SNIC) at multiple supercomputing centers (NSC, HPC2N, UPPMAX), partially funded by the Swedish Research Council through grant agreement no. 2016-07213.

**Competing interest.** The authors declare none.

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
