## [Reviewer Report]

This is an interesting paper detailing a case where it seems the rate of enzymatic reactions is largely controlled by the rate of protein dynamics - in this case loop motion. Energetic calculations and experimental measures of barriers for both a wildtype and a mutant chimera show similar values, but there are wildly different turnover rates. This is even more extreme in the case of one chimera that is totally inactive, but shows a chemical barrier of the same height as the wild type. This is interpreted to mean that once closed the potential energy surface is the same in all 3 cases, but in the case of the chimeras this loop closure happens far less frequently. This is one of those rare cases where potential energy calculations are of significant value in a complex system - clearly the free energy barrier to reaction must follow the rates. I think the paper should be published as is.

---

## [Reviewer Report]

This is a very nice and important contribution on loop dynamics in protein tyrosine phosphatases (PTP), which are crucial regulators of cellular signaling. The authors study PTP WT and two loop-chimeric constructs. These are important enzyme variants which can shed light on the WPD-loop motion. This loop motion optimally positions a catalytically critical residue into the active site and is directly linked to the turnover number of these enzymes.

The authors employ EVB simulations to study the effect of the loop variants on the reaction free energy of the rate limiting step of the reaction. Further, the authors employ pulling MD simulations and standard MD simulations to elucidate the dynamics of the WPD loop. The authors conclude that loop dynamics is likely the source of differences in catalytic rates in WT and chimeras. This work employs state-of-the-art and highly suitable computational tools, and the work is rigorously carried out. The manuscript is clear and concise. I recommend publication after attending to the following relatively minor points.

1. The authors find that the barriers for the second and rate-limiting step are similar in WT and chimeras 2 and 3. Based on this they conclude that the difference is due to loop dynamics and not chemistry. Did they also consider the possibility of chemistry taking place with only partially closed active site? Performing chemistry with a sub-optimally closed active site will likely require addition reorganizing of the loop during the chemical step that could be seen in increased activation barriers.

2. What was used as reaction coordinate during the pulling simulations?

3. Do the authors have any quantitative estimates of the relative differences in stability of the different loop states (closed, open, wide-open)? Did the authors consider methods like metadynamics to obtain such estimates?

4. When comparing the Ca RMSD in Figures S1 and S2, the small values during the EVB simulations are striking. Where some restraints applied during these simulations?

5. Do the authors have a quantitative means of discerning between the closed, open and wide-open protein conformations (i.e., some reaction coordinate)?

6. Some additional minor points:

a. “not chemistry upon creatin of these chimeric proteins.”  “not chemistry upon creation of these chimeric proteins.”

b. “our simulations of the wide-open state (Figure 4C) did however clear differences in the relative stabilities of for the different enzymes.”  ”our simulations of the wide-open state (Figure 4C) did however show clear differences in the relative stabilities of for the different enzymes.”

c. “starting from a different conformations of the WPD-loop,”  “starting from a different conformation of the WPD-loop,”